Production of fatty acids in Ralstonia eutropha H16 by engineering β-oxidation and carbon storage

Chen Janice S. 1 3
Colón Brendan 2
Dusel Brendon 1
Ziesack Marika 2
Way Jeffrey C. 1 Jeff.Way@wyss.harvard.edu
Torella Joseph P. 2
1 Wyss Institute for Biologically Inspired Engineering, Harvard Medical School , Boston, MA , United States
2 Department of Systems Biology, Harvard Medical School , Boston, MA , United States
3 Current affiliation: Department of Molecular and Cell Biology, University of California Berkeley , Berkeley, CA , United States
Chen Guo-Qiang
Electronic publication date: 2015 Dec 7
Publication date: 2015
Volume: 3
Electronic Location ID: e1468
Received 2015 Aug 20; Accepted 2015 Nov 12
Copyright: © 2015 Chen et al.
Copyright year: 2015
Copyright holder: Chen et al.
License: This is an open access article distributed under the terms of the Creative Commons Attribution License, which permits unrestricted use, distribution, reproduction and adaptation in any medium and for any purpose provided that it is properly attributed. For attribution, the original author(s), title, publication source (PeerJ) and either DOI or URL of the article must be cited.
License URL: https://creativecommons.org/licenses/by/4.0/

Keywords: -oxidation, Acyl-CoA ligase, Ralstonia, Biofuel, Metabolic engineering

Funding: ONR Multidisciplinary University Research Initiative N00014-11-1-0725 National Science Foundation Graduate Research Fellowship This work was supported by the ONR Multidisciplinary University Research Initiative (MURI) Award N00014-11-1-0725 (PAS), and a National Science Foundation Graduate Research Fellowship (JPT). The funders had no role in study design, data collection and analysis, decision to publish, or preparation of the manuscript.

==============================
Ralstonia eutropha H16 is a facultatively autotrophic hydrogen-oxidizing bacterium capable of producing polyhydroxybutyrate (PHB)-based bioplastics. As PHB’s physical properties may be improved by incorporation of medium-chain-length fatty acids (MCFAs), and MCFAs are valuable on their own as fuel and chemical intermediates, we engineered R. eutropha for MCFA production. Expression of UcFatB2, a medium-chain-length-specific acyl-ACP thioesterase, resulted in production of 14 mg/L laurate in wild-type R. eutropha. Total fatty acid production (22 mg/L) could be increased up to 2.5-fold by knocking out PHB synthesis, a major sink for acetyl-CoA, or by knocking out the acyl-CoA ligase fadD3, an entry point for fatty acids into β-oxidation. As ΔfadD3 mutants still consumed laurate, and because the R. eutropha genome is predicted to encode over 50 acyl-CoA ligases, we employed RNA-Seq to identify acyl-CoA ligases upregulated during growth on laurate. Knockouts of the three most highly upregulated acyl-CoA ligases increased fatty acid yield significantly, with one strain (ΔA2794) producing up to 62 mg/L free fatty acid. This study demonstrates that homologous β-oxidation systems can be rationally engineered to enhance fatty acid production, a strategy that may be employed to increase yield for a range of fuels, chemicals, and PHB derivatives in R. eutropha.

Introduction

Ralstonia eutropha H16 is a facultatively chemolithoautotrophic bacterium capable of using H2 and CO2 as sole sources of energy and carbon, respectively (Pohlmann et al., 2006). Under conditions of carbon abundance and nutrient deprivation, R. eutropha accumulates polyhydroxybutyrate (PHB), an intracellular carbon and energy storage polymer (Verlinden et al., 2007). The naturally high production rate and yield of PHB and PHB-derived bioplastics, coupled with the industrial scalability and facile genetics of R. eutropha, have made it an attractive host for biofuel and industrial chemical production (Henderson & Jones, 1997). Recently, it has been shown that through the expression of heterologous enzymes, precursors in PHB synthesis can be redirected to the synthesis of short-chain fusel alcohols such as isopropanol and isobutanol (Grousseau et al., 2014; Li et al., 2012; Lu et al., 2012). This work has demonstrated the plausibility of using R. eutropha as a chassis for fuel and chemical production beyond PHB-based bioplastics.

The capacity of R. eutropha to grow autotrophically using CO2 and H2/formate has also made it an attractive chassis organism for “electrosynthesis”: the use of electricity to drive biological carbon fixation for renewable chemical production (Nevin et al., 2010; Rabaey et al., 2010), an alternative to non-renewable oil-based methods. Owing to the difficulty of electrochemically reducing CO2 to infrastructure-compatible carbon-based fuels (Kuhl et al., 2012), and the forty-year-old knowledge that R. eutropha is capable of robust growth in actively electrolyzed media (Schlegel & Lafferty, 1965), electrosynthesis using R. eutropha has emerged as an attractive future technology for renewable fuel and chemical production. Efforts by Li and colleagues (2012) demonstrated the ability to integrate precious metal-based electrochemical formate production with engineered R. eutropha to convert electricity and CO2 to isobutanol and 3-methyl-1-butanol. In recent work from our lab, we coupled R. eutropha with scalable, earth-abundant electrodes to produce isopropanol at high efficiency, and achieved biomass yields comparable to those of terrestrial plants (Torella et al., 2015). These studies provide a proof of principle demonstration that R. eutropha may be useful for the renewable production of fuels and chemicals from electricity and CO2.

While many efforts to engineer R. eutropha have focused on the production of bioplastics or short-chain alcohols by redirecting the PHB synthesis pathway, few have focused on engineering more chemically diverse products for fuel and chemical applications. Fatty acids represent a broad category of important chemicals that are facile to produce from precursors in bacterial metabolism (Fig. 1), and may be converted biologically or chemically into a range of biofuels and valuable industrial chemicals including alcohols, alkanes, and wax esters (Steen et al., 2010; Lennen & Pfleger, 2013; Akhtar, Turner & Jones, 2013). Moreover, medium-chain fatty acids (MCFAs) may be incorporated into the PHB production pathway to make copolymers with superior physical properties compared to pure PHB (Lageveen et al., 1988).

Figure 1 Schematic of fatty acid production and uptake by Ralstonia eutropha H16.

Heterotrophic and autotrophic growth of R. eutropha H16 both result in production of the central metabolite acetyl-CoA (bold arrow). During fatty acid synthesis, derivatives of acetyl-CoA are iteratively condensed on an acyl carrier protein (ACP) scaffold to yield a series of acyl-ACPs of increasing length for lipid synthesis. Heterologous expression of chain-length-specific acyl-ACP thioesterases (tes) enables the hydrolysis of acyl-ACPs and release of fatty acids, some of which are capable of crossing the lipid membrane. Fatty acids can be re-consumed through the β-oxidation pathway, which iteratively removes two carbons from the fatty acid to yield acetyl-CoA (top dashed line). Entry of fatty acids into β-oxidation is catalyzed by one or more acyl-CoA ligases (fadD homologues). Multiple homologues of each step of the β-oxidation pathway have been identified in the genome of R. eutropha H16, with the number of homologues indicated in parentheses. Acetyl-CoA may also be converted to polyhydroxybutyrate (PHB) through the action of three enzymes (phaCAB) under periods of carbon excess but limitation of nutrients such as nitrogen or phosphate. β-oxidation and PHB synthesis share common metabolic intermediates and may be co-regulated. The dashed line indicates the potential for 3-hydroxyl-acyl-CoAs to enter PHB synthesis pathway and be incorporated into the polymer.

Recent efforts to produce fatty acids and fatty acid-related chemicals in R. eutropha include the successful production of long-chain fatty acids and methyl ketones (Muller et al., 2013), as well as alcohols, aldehydes and alkanes (Bi et al., 2013). While significant yields of methyl ketones have been achieved, the fatty acid precursors to ketones have only been produced at a maximum yield of 0.017 mg/L in 2% fructose minimal medium and 10 mg/L in LB medium (Muller et al., 2013). This limitation is likely due to extensive fatty acid degradation, or β-oxidation. R. eutropha expresses over 50 homologues of some of its β-oxidation enzymes, far exceeding that of other model organisms such as Escherichia coli, which contains only one or two homologues for each enzyme in its β-oxidation pathway (Shimizu et al., 2013) (Fig. 1). This apparent redundancy may pose a significant challenge to engineering the production of fatty acids and related chemicals in R. eutropha.

In this work, we engineered R. eutropha to produce the medium-chain fatty acid laurate by expressing UcFatB2, a medium-chain-length-specific acyl-ACP thioesterase, and obtained laurate yields of up to 14 mg/L. Fatty acid yields were increased by deleting one of several acyl-CoA ligases (fadD3, A0285, A2794) that catalyze the first step in fatty acid β-oxidation. These were identified by RNA-Seq as genes strongly upregulated in R. eutropha during growth on lauric acid. Fatty acid yield could also be increased by knocking out PHB synthesis (phaCAB). In all, we achieved maximum production of 62 ± 2 mg/L free fatty acids, the highest yield of fatty acids reported in this organism to date. This work provides a starting point for the production of fatty acid-derived fuels and chemicals at high yield, including biofuels and PHB-derived bioplastics, in R. eutropha.

Materials and Methods

Bacterial growth and induction

Strains and plasmids used in this study are listed in Table S1. E. coli strains were grown at 37 °C in lysogeny broth (LB) supplemented with 50 µg/ml kanamycin as appropriate. R. eutropha strains were grown at 30 °C in rich medium (16 g/L nutrient broth, 10 g/L yeast extract, 5 g/L (NH4)2SO4) + 10 µg/ml gentamicin, and supplemented with 200 µg/ml kanamycin as appropriate. Rich medium plates were prepared from the same media plus 1.5% agar (BD). Lauric acid minimal media plates were produced with 1.5% agar and the following salts: 4.0 g/L NaH2PO4, 4.6 g/L Na2HPO4, 0.45 g/L K2SO4, 0.39 g/L MgSO4, 0.062 g/L CaCl2, 0.05% (w/v) NH4Cl, 0.1% lauric acid, 1% NP40 and 1 ml/L of a trace metal solution (15 g/L FeSO4 ⋅ 7H2O, 2.4 g/L MnSO4 ⋅ H2O, 2.4 g/L ZnSO4 ⋅ 7H2O, and 0.48 g/L CuSO4 ⋅ 5H2O in 0.1 M HCl). For lauric acid minimal media plate growth experiments, R. eutropha strains were streaked out onto rich medium plates for 2 days at 30 °C, and then a single colony was transferred onto minimal media plates containing 0.1% laurate and grown at 30 °C for 5 days before imaging with a Canon EOS Rebel T3i Digital SLR camera.

To generate liquid cultures of E. coli or R. eutropha, a single freshly-grown colony was inoculated into 10 mL of culture medium in a 14 mL BD Falcon disposable polypropylene culture tube and shaken at 220 rpm overnight. R. eutropha free fatty acid (FFA) production experiments were conducted in 2 mL 96-well plates (Thermo Scientific, Waltham, MA, USA) or 30 mL glass culture tubes. In both cases, individual R. eutropha colonies were first inoculated in 10 mL rich medium with appropriate antibiotics and grown at 220 rpm at 30 °C for 20–22 h. For FFA production in 96-well plates, R. eutropha overnight cultures were diluted 1:20 into 1 mL fresh rich medium plus necessary antibiotics and shaken at 30 °C, 1,200 rpm on a Titramax 1,000 platform shaker (Heidolph, Schwabach, Germany) for 4–5 h before induction with 0.5% L-arabinose. For FFA production in glass culture tubes, R. eutropha overnight cultures were diluted 1:20 into 1 ml fresh rich medium supplemented with required antibiotics and shaken at 180 RPM at 30 °C. After 20 h, these cultures were induced with 0.5% L-arabinose and a 20% (v/v) dodecane overlay added. OD600 was measured using an Ultrospec 10 spectrophotometer (Amersham, Little Chalfont, UK).

Construction of plasmids

All L-arabinose-inducible plasmids were generated by conventional restriction cloning. L-arabinose-inducible mCherry was digested from pJT390 (constructed by Joseph P. Torella) with SpeI and NaeI and ligated into SpeI and EcoRV sites in pBBR1MCS2 to generate pJC007. C-terminally his-tagged UcFatB2 (codon-optimized for expression in Synechococcus elongatus) was PCR amplified from pJT204 (constructed by Joseph P. Torella) using forward primers JC016–JC019 and 5′-phosphorylated reverse primer JC023 to generate four RBS variants driving UcFatB2. The resulting PCR products were digested with XbaI and inserted into XbaI and AleI sites in pJC007 to construct plasmids pJC022–pJC025. All markerless deletion plasmids were generated by conventional restriction cloning or Gibson assembly. A gene deletion fragment consisting of 240 bp stretches directly upstream and downstream the A0285 open reading frame was linked by a PacI restriction site, and the 498 bp product with flanking BamHI sites was synthesized directly by Integrated DNA Technologies (IDT). The synthesized insert and backbone vector pCB96 was digested with the double-cutter BamHI, and ligated to generate the A0285 markerless deletion plasmid, pJC043. The B1148 markerless deletion plasmid was generated as described by Lu et al. (2012), using primer pairs pJC078/JC079 and pJC080/JC081 for overlap extension PCR. The resulting PCR product and parent plasmid pCB96 were digested with XbaI and SacI and ligated to generate pJC046. The gene deletion fragment for A2794 was synthesized by IDT as described above, except the fragment was flanked upstream by XbaI and downstream by SacI, with ∼30 bp overlap regions to pCB96 on both ends. The fragment was assembled to the pCB96 backbone (linearlized by XbaI/SacI) via Gibson Assembly to generate the A2794 markerless deletion plasmid, pJC045. All newly-ligated plasmids and Gibson assembly reactions were transformed into TSS-competent E. coli TURBO or Mach1 cells, isolated and re-transformed into E. coli S17-1 cells for conjugative transfer of plasmid DNA into R. eutropha. Following conjugation, UcFatB2 thioesterase plasmids (pJC022–pJC025) were extracted from R. eutropha and frozen for subsequent electroporation. Primers used in plasmid construction are listed in Table S2.

Genomic modifications

Genetic knockouts were constructed as described previously in Brigham et al. (2010). R. eutropha was conjugated with a donor E. coli strain S17-1 harboring a markerless gene deletion plasmid containing homology to the 250 bp regions upstream and downstream of the gene of interest, and containing a kanamycin resistance cassette and sacB gene for sucrose-based counter-selection. To perform the mating procedure, 10 ml of R. eutropha and E. coli were grown separately in rich medium at 30 °C and LB at 37 °C, respectively. Each culture was harvested after 20 h of shaking, washed twice and resuspended in 1 ml 0.85% NaCl. Next, 30 µl of resuspended R. eutropha was mixed with 20 µl of resuspended E. coli and the 50 µl mixture plated onto an LB plate without selection for 24 h at 30 °C. Following conjugation, a cell scraper was used to remove half the cells from the plate, which were then transferred to an eppendorf tube and resuspended in 300 µl 0.85% NaCl. To select for R. eutropha with successful plasmid integration, either 100 µL or 10 µL of the cell suspension was plated onto rich medium agar plates with 10 µg/ml gentamicin and 200 µg/ml kanamycin and grown for 48 h at 30 °C. R. eutropha colonies were picked and restreaked onto rich medium plates supplemented with gentamicin and kanamycin and grown for 24 h at 30 °C. To allow the second recombination event to occur, the restreaked colonies were grown in 10 mL rich medium with 10 µg/ml gentamicin at 30 °C. After 20 h, 1 ml from the overnight culture was harvested, washed twice with 0.85% NaCl and resuspended in 1 ml 0.85% NaCl. 10 µl of resuspended cells were then plated onto rich medium plates containing 10 µg/ml gentamicin and 5% sucrose. The plates were incubated at 30 °C for 36–48 h, and colonies screened with colony PCR for successful recombination-based loss of the gene of interest. Primers used in strain verification are listed in Table S2. Note that although the B1148 markerless deletion plasmid was constructed, we were unable to knock out the B1148 gene in parallel with other fadD homologues identified by RNA-Seq. The lack of viable colonies from multiple attempts to knock out B1148 in WT and ΔfadD3 background strains may reflect that the gene is essential, or be a consequence of the locally high GC content of the genome.

Generation of R. eutropha MCFA production strains

Freshly-prepared electrocompetent R. eutropha strains were generated for all FFA production experiments using UcFatB2 thioesterase plasmids (pJC022–pJC025). Single R. eutropha colonies were grown overnight in rich medium, diluted 1:10 in 10 ml rich medium in 15 ml falcon tubes, and harvested at OD600 = 0.8. Cultures were submerged in ice for 10 min, pelleted and washed twice with 1 ml ice-cold 10% sucrose. The cell pellet was resuspended in 1 ml 10% sucrose, and 50 ul electrocompetent cells were combined with 1–2 µg of plasmid DNA. The cell suspension was transferred to a pre-chilled 0.1 cm gap cuvette and electroporated using the Biorad electroporator manually set at 1.15 kV. Immediately following electroporation, 250 µl rich medium was added and the cells were transferred to an eppendorf tube. Electroporated cells were recovered at 30 °C for 2 h before being plated on selective medium and grown for 2–3 days at 30 °C.

Enzymatic assays for FFA measurements

Fatty acids were measured using the enzymatic Free Fatty Acid Half Micro Test (Roche, Basel, Switzerland) as described by Torella et al. (2013). For FFA measurements with 20% dodecane, 5 µl of the overlay was diluted into 20 ul dodecane before assaying with the same sample volume as previously discussed. Because acyl-CoAs are not cell permeable, detection of these compounds by the enzymatic assay is not possible in the cell supernatant or dodecane overlay.

GCMS identification of lauric acid

Fatty acids were extracted from 400 µl of acidified culture with ethyl acetate and esterified with ethanol. Fatty ethyl esters were extracted with hexane and run on an Agilent GC-MS 5975/7890 (Agilent Technologies, Santa Clara, CA, USA) using an HP-5MS (length: 30 m; diameter: either 0.25 or 0.50 mm; film: 0.25 µm) column, as described by Torella et al. (2013). This method is unable to detect acyl-CoAs, because they cannot be extracted into the solvent layer due to their hydrophilic properties.

Isolation of RNA and RNAseq analysis

For RNA-Seq experiments, individual colonies of R. eutropha strain Re2061 (ΔphaCAB) were inoculated in triplicate into rich medium, grown overnight, then diluted 1:100 into 30 mL minimal medium (Brigham et al., 2010) in a 125 mL flask, supplemented with 2.0% Nondiet-P40 (v/v), 0.54% NH4Cl, 10 µg/ml gentamicin, and either 0.4% fructose (w/v) or 0.1% lauric acid (w/v). Cells were grown to stationary phase, diluted 1:100 again into the same medium, and each culture monitored at 2 h intervals for cell density by OD600. Once a given culture entered mid-log phase (OD600 = 0.5–0.6), 1 mL was immediately taken and stabilized with RNA Protect (Qiagen, Hilden, Germany), followed by purification with an RNEasy Mini Kit (Qiagen, Hilden, Germany). The purified RNA was then analyzed by Genewiz on an Illumina HiSeq-2500, and the resulting data analyzed with Rockhopper (McClure et al., 2013) for alignment to the R. eutropha H16 genome and differential gene expression analysis.

Results

Expression of the medium-chain thioesterase UcFatB2 enables MCFA production in R. eutropha

To produce the MCFA lauric acid, we cloned UcFatB2, a thioesterase selective for 12-carbon acyl-ACP substrates from the plant U. californica (Voelker & Davies, 1994), into an arabinose-inducible, R. eutropha-compatible replicative vector (pJC007; Table S1). Ribosome Binding Sites (RBS) with a range of predicted expression strengths (Translational Initiation Region (TIR) values from 14,000 to 84,000 in R. eutropha) were designed using the RBS Calculator (Salis, Mirsky & Voigt, 2009) and cloned upstream of UcFatB2 (pJC022–pJC025; Table S1). One variant (pJC024, TIR = 66,000) resulted in 22 mg/L total free fatty acid (FFA) production within 48 h (Fig. 2A). The absence of detectable fatty acid production in the variant with the highest predicted TIR (84,000, pJC025) may reflect toxicity or inclusion body formation resulting from high expression, or insufficient protein production (e.g., in the case that the actual TIR is much lower than predicted).

Figure 2 Production of lauric acid by engineered R. eutropha.

(A) Total free fatty acids (FFA) produced by ReH16 grown on rich broth (RB) and expressing UcFatB2 from plasmids (top row) of different predicted RBS strengths (RBS Calculator; bottom row). FFAs were detected in culture supernatant 48 h after arabinose induction by the fatty acid half-micro assay (Methods). Negligible amounts of free fatty acid (0.04 ± 0.06 mg/L) were detected for ReH16 expressing pJC025. (B). Chain length distribution of saturated fatty acids extracted from total culture of ReH16-pJC024 at 48 h post-induction via GCMS. For all experiments in this figure, error bars represent the standard error of the mean (S.E.M.) from N ≥ 3 independent experiments.

GCMS analysis of the ReH16-pJC024 culture confirmed production of lauric acid at 14.3 ± 2.6 mg/L after 24 h of induction; laurate production did not occur in wild-type ReH16 (Fig. 2B). Compared to lauric acid, significantly smaller amounts of myristic (C14:0) and palmitic (C16:0) acids were detected from both wild-type and UcFatB2-expressing strains (Fig. 2B, ∼2 mg/L), while the concentration of shorter and unsaturated products was negligible.

Although FFA production was significant within the first ∼24 h following arabinose induction, by 50 h post-induction FFAs were undetectable in the culture (Fig. 3A). This behavior is consistent with the expectation that fatty acids should be re-consumed by R. eutropha through β-oxidation, something previously observed in other microbes engineered to produce fatty acids (Lu, Vora & Khosla, 2008; Runguphan & Keasling, 2014).

Figure 3 β-oxidation limits fatty acid yield in engineered R. eutropha.

(A) Time course of free fatty acid (FFA) production by different R. eutropha mutants in RB following arabinose induction (0 h time point). (B) Total FFA production at 24 h as a function of genetic background and UcFatB2 expression plasmid; each plasmid has a different RBS and predicted expression strength. (C) Time course of fatty acid uptake by R. eutropha of different genetic backgrounds incubated with 120 mg/L lauric acid in RB. For all experiments in this figure, FFA concentration was measured using the fatty acid half-micro assay (Methods), and error bars represent S.E.M. for N ≥ 3 independent experiments.

Fatty acid production in R. eutropha is limited by β-oxidation and PHB synthesis

To evaluate the role of β-oxidation in FFA reconsumption, we expressed UcFatB2 in R. eutropha strain Re2303, which contains knockouts in two putative β-oxidation operons and is incapable of growth on oleic acid (Brigham et al., 2010). In contrast to the wild-type, Re2303 (Δβ-oxidation) produced FFA concentrations up to 20 mg/L without reconsumption over a 150 h time course, indicating that β-oxidation in the wild-type contributes to the decline in FFA yields (Fig. 3A).

Although Re2303 lacks two β-oxidation operons, the acyl-CoA ligases that catalyze entry of fatty acids into the β-oxidation pathway are not located in these operons. We therefore hypothesized that FFA yield might be limited by uptake of fatty acids and conversion to acyl-CoAs and other β-oxidation intermediates, which are not detected in either of our fatty acid assays (Methods). We assayed FFA production in Re2312, which contains a deletion in a known acyl-CoA ligase in R. eutropha, fadD3 (Brigham et al., 2010), and observed a burst of 30 mg/L FFAs within 4h post-induction along with a steady increase up to 54 mg/L FFA over the following week. By 150 h post-induction, Re2312 (ΔfadD3) led to a 2.8-fold improvement over the Re2303 (Δβ-oxidation) mutant strain and an 8.5-fold increase over wild-type parent strain (Figs. 3A and 3B), indicating that disruption of the first step in β-oxidation enhanced the production of FFAs beyond what is achieved by blocking downstream steps in β-oxidation (e.g., in Re2303).

As PHB synthesis is a sink both for acetyl-CoA (Cook & Schlegel, 1978) and β-oxidation intermediates (Shimizu et al., 2013) in R. eutropha, we hypothesized that it may also limit FFA yield. Expression of UcFatB2 in a phaCAB deletion strain (Re2061) resulted in a considerable 5-fold increase in FFA yield over wild-type by 150 h. Unlike the initial burst of FFAs observed in Re2312 (ΔfadD3), the increase in FFA production by Re2061 (ΔphaCAB) began 40 h after arabinose induction, suggesting a different mechanism for increasing FFA yield. Indeed, PHB synthesis is induced in response to nutrient limitation (Verlinden et al., 2007), which may explain the increase in FFA synthesis at a late time-point in this strain (Figs. 3A and 3B). It is worth noting that simultaneously knocking out PHB synthesis and beta-oxidation genes was unable to enhance FFA production over that achieved by ΔfadD3 alone (Fig. S1), suggesting that beta oxidation is the limiting factor for FFA yield. Our study therefore focused on manipulating the acyl-CoA ligases responsible for lauric acid consumption to enhance fatty acid yields in R. eutropha.

Consistent with the hypothesis that Re2312 (ΔfadD3) had increased FFA production due to a lower rate of FFA consumption, we found that it had a slower rate of exogenous FFA uptake (Fig. 3C). Wild-type R. eutropha, Re2303, Re2061 and Re2312 (none expressing UcFatB2) were incubated with 120 mg/L laurate and the levels of extracellular FFA monitored over a 24 h time period. The wild-type, Re2303 (Δβ-oxidation) and Re2061 (ΔphaCAB) strains had similar consumption rates with laurate no longer detectable in the media by 12 h. Re2312 (ΔfadD3) cells exhibited a slower rate of consumption relative to the other strains, with laurate becoming undetectable by 24 h (Fig. 3C). Laurate consumption was not blocked entirely, suggesting that other acyl-CoA ligases may still be active in Re2312. Indeed, we found that Re2312 was still capable of growth on laurate minimal medium (Fig. S2).

Identification of three additional acyl-CoA ligases that limit FFA yield through RNA-Seq

Deletion of the primary acyl-CoA ligase in R. eutropha, fadD3, results in slower uptake of fatty acids (Fig. 3C) but does not block their utilization as a sole carbon source (Brigham et al., 2010). This is consistent with computational predictions that the R. eutropha genome harbors an apparently redundant β-oxidation system (Pohlmann et al., 2006), estimated to contain 51 homologues of the acyl-CoA ligase alone (Shimizu et al., 2013) (Fig. 4A).

Figure 4 Three additional acyl-CoA ligases limit FFA yield.

(A) Schematic diagram of β-oxidation in R. eutropha. The first enzymatic step of β-oxidation is carried out by an acyl-CoA ligase (ACL) followed by an acyl-CoA dehydrogenase (ACD), enoyl-CoA hydratase (ECH), 3-hydroxyacyl-CoA dehydrogenase (HCD) and β-ketoacyl-CoA thiolase (BKT). Each β-oxidation gene has multiple annotated homologues in the R. eutropha genome (number indicated in parentheses). (B) Time course of Re2061 growth on fructose or laurate minimal medium. Error bars represent S.E.M. from N = 3 independent experiments. The grey arrow indicates the point at which cultures were sampled for RNA-Seq analysis. (C) Expression levels of four acyl-CoA ligases upregulated during growth on laurate, measured by RNA-Seq and normalized by the upper quartile method (Methods). Numbers above bars indicate fold-increase in laurate media as compared to fructose media. Error bars represent S.E.M. of the normalized expression levels for three independent experiments. Differences in expression between fructose and laurate conditions are significant for all four genes after correcting for multiple hypothesis testing (q-value < 0.001). (D) Total FFA production 96 h after arabinose induction by R. eutropha knockout mutants, as measured by the fatty acid half-micro assay (top). OD600 96 h after induction for each mutant. Error bars represent S.E.M. from N = 4 independent experiments in both panels. ∗ indicates a significant difference in mean FFA production with p < 0.05, evaluated using the one-tailed student’s t-test.

To narrow down the list of acyl-CoA ligases likely to limit the production of fatty acids, a preliminary RNA-Seq experiment was performed to identify acyl-CoA ligases that are up-regulated during growth on lauric acid. Three independent cultures of Re2061 were grown from an OD600 of 0.02 in minimal medium containing fructose or lauric acid as a sole carbon source (Fig. 4B). During mid-exponential phase (OD 0.3–0.6), a sample of each culture was taken for analysis by RNA-Seq (Methods). Successfully aligned transcript frequency was low (∼40–50% of reads), likely due to poor sample quality, and we therefore elected not to analyze the resulting data in full here (a future manuscript is forthcoming). Nevertheless, it provided enough information on acyl-CoA ligase expression patterns to engineer improved FFA production in R. eutropha.

Read counts normalized to upper-quartile gene expression (McClure et al., 2013; Bullard et al., 2010) were used to identify four genes annotated as acyl-CoA ligases in the R. eutropha genome that were upregulated strongly (≥5-fold) and significantly (q-value < 0.001) during growth on laurate as compared to fructose: fadD3, A2794, A0285, and B1148 (Fig. 4C). We generated mutants of each gene alone and in combination with fadD3 with the exception of B1148, which we were unable to knock out after several attempts (Methods). FFA production and OD600 were measured for each strain 96 h after induction; a large increase in FFA production was associated with knocking out fadD3 (Re2312), A0285 (JC357) and A2794 (JT338), with A2794 providing the greatest FFA yield at 62 ± 2 mg/L (Fig. 4D). The ΔA0285ΔfadD3 double knockout (JC358) failed to increase FFA yield significantly above what was achieved for either knockout alone, while the ΔA2794ΔfadD3 double knockout (JT339) actually decreased FFA yield. As the latter was the only strain tested with impaired biomass yield (Fig. 4D, bottom panel), this reduced mean yield may reflect a synthetic loss of fitness associated with the ΔA2794ΔfadD3 double knockout.

Discussion

R. eutropha is a versatile industrial organism capable of synthesizing a range of biofuels and bioplastics either heterotrophically from organic substances or autotrophically from H2 and CO2. In this study, we engineered medium-chain fatty acid (MCFA) synthesis in R. eutropha by optimizing the expression of a heterologously expressed thioesterase, UcFatB2. Deletion of the acyl-CoA ligase fadD3 increased fatty acid yields without completely abolishing fatty acid uptake, implying the action of additional acyl-CoA ligases. Because 51 such enzymes have been annotated in the R. eutropha genome, we employed RNA-Seq to identify which were upregulated during growth on the MCFA lauric acid. Of the four candidate genes identified, knockouts in three (fadD3, A0285, A2794) led to higher fatty acid yields, with one (ΔA2794) providing the highest yield of fatty acids yet reported in R. eutropha (62 mg/L).

Expression of UcFatB2 in R. eutropha resulted in relatively selective production of lauric acid, as laurate production increased by 15 mg/L, while all other detected fatty acids increased by less than 1 mg/L (Fig. 2B). This level of selectivity appears higher than in E. coli, where expression of UcFatB2 has previously been shown to liberate a significant amount of decanoate, myristate and palmitate (Torella et al., 2013). Likewise, although UcFatB2 has previously been shown to produce methyl ketones in E. coli (presumably by catalyzing hydrolysis of β-ketoacyl-ACPs (Voelker & Davies, 1994)), we failed to detect methyl ketones in R. eutropha expressing UcFatB2 by GCMS. These results may suggest different rates of product degradation, a difference in interaction between UcFatB2 and R. eutropha ACP, or a distinct distribution of acyl- and β-ketoacyl-ACPs in R. eutropha that result in altered product profiles.

During our fatty acid production timecourse, the ΔfadD3 strain produced an initial burst of 30 mg/ml free fatty acids in the 4 h following thioesterase induction that was not observed in the wild-type (Fig. 3A). These kinetics suggest that the rate of acyl-ACP hydrolysis by UcFatB2 is faster than the rate at which free fatty acids leave the cell, and that the intracellular pool of free fatty acids are rapidly acted on by FadD3 in the wild-type to generate acyl-CoAs. This hypothesis is consistent with a previous observation by Muller et al. (2013) in which R. eutropha expressing a thioesterase in a Δβ-oxidation background produced free fatty acids that accounted for only ∼0.00016% of the fructose carbon consumed. However, addition of a heterologous methyl ketone pathway that acts on β-oxidation intermediates produced greater than 50 mg/L methyl ketones, or about 2% of the fructose carbon consumed (Muller et al., 2013).

Unlike in the ΔfadD3 strain, in the ΔphaCAB strain the largest increase in fatty acid production began at a ∼30 h delay post-induction (Fig. 3A). This difference in timing may reflect a difference in mechanism for increasing fatty acid yields. Under conditions of nutrient deprivation, WT R. eutropha normally induce PHB synthesis (Wang & Yu, 2007); phaCAB strains, however, accumulate acetyl-CoA, which is converted to pyruvate or can be diverted by engineered pathways to other products (e.g., isopropanol or isobutanol (Grousseau et al., 2014; Li et al., 2012; Lu et al., 2012)). High acetyl-CoA levels stimulate fatty acid synthesis and inhibit beta-oxidation (Torella et al., 2013; Dellomonaco et al., 2011), and could explain the increase in fatty acid production observed in the ΔphaCAB mutant. Moreover, as acetyl-CoA would only accumulate after nutrient depletion, the observed ∼30 h delay may reflect the time required to deplete an essential nutrient in the growth medium.

Our identification of differential expression of fadD homologs (fadD3, A0285, A2794 and B1148) in R. eutropha during growth on lauric acid and fructose using RNA-Seq (Fig. 4C) was consistent with a previously published microarray study (Brigham et al., 2010), and hints at the possibility of differential regulation of β-oxidation enzymes in response to fatty acids of different chain lengths. Previous work by Brigham et al. (2010) identified significant upregulation of fadD3 (A3288) and A2794 during growth on trioleate relative to fructose, similar to our observation that these genes are upregulated during growth on laurate. However, they did not observe increased expression of A0285 or B1148, though these were strongly upregulated in response to laurate in our system. These results may indicate that different β-oxidation enzymes in R. eutropha are induced by structurally distinct fatty acid substrates, discriminating for example on the basis of saturation and chain length.

Knockouts of fadD3, A0285 and A2794 expressing UcFatB2 increased FFA production by 8.9, 10.2 and 12.5-fold (respectively) over WT at 96 h post-induction, without altering total biomass yield (Fig. 4D). Surprisingly, knocking out candidate acyl-CoA ligases in conjunction with fadD3 failed to further improve yield: the ΔA0285ΔfadD3 double mutant did not yield any added FFA increase over the individual mutations, and the ΔA2794ΔfadD3 double mutant had lower yields than either of the single mutants alone. The failure to increase yield in these double mutants may be due to compensatory regulation of acyl-CoA ligases in response to an increase in laurate levels, to kinetic features of R. eutropha’s extensive β-oxidation system, or both. In the specific case of ΔA2794ΔfadD3, the drop in productivity may also be due to a fitness defect, indicated by the relative decrease in total biomass yield at 96 h compared to its parent strains (Fig. 4D). As FadD is essential for stationary phase survival in E. coli (Farewell et al., 1996), it is possible that acyl-CoA ligases in R. eutropha perform compensatory roles in the maintenance of stationary phase fitness, and that the deletion of some combinations of ligases carry a stationary phase fitness cost. In all, these results indicate a potentially complex metabolic and regulatory relationship among the acyl-CoA ligases in R. eutropha that may be addressed through more systematic engineering, as well as transcriptomic and metabolic investigation.

Although the roles of many of the unannotated β-oxidation homologs remain elusive, identifying transcriptomic changes in response to a range of fatty acid chain lengths may help elucidate these functions, while providing the information necessary to engineer R. eutropha for industrially relevant FFA yields. It will also be important to better understand the relationship between PHB synthesis and fatty acid metabolism, both because their co-regulation complicates efforts to produce FFAs at high yields, and because of the importance of R. eutropha as a chassis for industrial polyhydroxyalkanoate (PHA) production. As PHA co-polymers containing medium-chain-length side chains have superior physical properties compared to PHB (Nomura et al., 2004) and these side chains are derived from β-oxidation intermediates (Tsuge et al., 2005; Insomphun et al., 2014), a desirable application of the present research will be to engineer R. eutropha for the production, partial β-oxidation, and incorporation of MCFAs into PHA co-polymers. In particular, the link between β-oxidation and PHA synthesis has been demonstrated in two related PHA-producing species, Pseudomonas putida and Pseudomonas entomophila: knocking out key enzymes in β-oxidation was sufficient to increase metabolic flux through the PHA synthesis pathway, and this enabled chain-length-specific production of PHA consisting of 3-hydroxydodecanoate when grown on lauric acid (Liu et al., 2011; Chung et al., 2011). We also note that while our study lays a foundation for improved FFA production by R. eutropha during growth in rich medium, it will be important to demonstrate high-yield production of FFAs in minimal carbon media and on H2 and CO2 in order to exploit the full capacity of this metabolically versatile, chemolithotrophic bacterium. In particular, H2 and CO2-based FFA production would enable integration with water splitting catalysts for efficient solar-to-fuel transformation (Li et al., 2012; Torella et al., 2015), offering new, sustainable routes for the production of chemicals, bioplastics, and fuels.

Supplemental Information

Supplemental Information 1 Production of lauric acid by engineered R. eutropha

Click here for additional data file.

Supplemental Information 2 β-oxidation limits fatty acid yield in engineered R. eutropha

Click here for additional data file.

Supplemental Information 3 Three additional acyl-CoA ligases limit FFA yield

Click here for additional data file.

Figure S1 Knocking out genes in β-oxidation and PHB synthesis simultaneously does not enhance FFA production

Total free fatty acids (FFA) produced by R. eutropha mutants expressing UcFatB2 in rich broth supernatant 24 h after arabinose induction. Error bars represent the standard error of the mean (S.E.M.) from N ≥ 3 independent experiments.

Click here for additional data file.

Supplemental Information 4 Knocking out genes in β-oxidation and PHB synthesis simultaneously does not enhance FFA production

Click here for additional data file.

Figure S2 ΔfadD3 is capable of growth on laurate minimal medium

Growth of WT ReH16 and three independent ΔadD3 clones on a (A) rich medium plate after 2 days or (B) minimal medium ±0.1% lauric acid plate after 5 days at 30 °C.

Click here for additional data file.

Table S1 Strains and plasmids list

Click here for additional data file.

Table S2 Primer list

Click here for additional data file.

We thank PA Silver for project guidance, J Lu and AJ Sinskey for plasmids and reagents; J Lu, CJ Brigham, TJ Ford, DC MacKellar and O Mavrothalassitis for helpful discussion.

Additional Information and Declarations

Competing Interests

Author Contributions

Data Availability

The authors declare there are no competing interests.

Janice S. Chen and Joseph P. Torella conceived and designed the experiments, performed the experiments, analyzed the data, contributed reagents/materials/analysis tools, wrote the paper, prepared figures and/or tables, reviewed drafts of the paper.

Brendan Colón and Marika Ziesack performed the experiments, analyzed the data, contributed reagents/materials/analysis tools, reviewed drafts of the paper.

Brendon Dusel performed the experiments, analyzed the data, contributed reagents/materials/analysis tools.

Jeffrey C. Way conceived and designed the experiments, analyzed the data.

The following information was supplied regarding data availability:

Fructose—replicate 1: https://zenodo.org/record/34443

Fructose—replicate 2: https://zenodo.org/record/34444

Fructose—replicate 3: https://zenodo.org/record/34445

Lauric acid—replicate 1: https://zenodo.org/record/34446

Lauric acid—replicate 2: https://zenodo.org/record/34447

Lauric acid—replicate 3: https://zenodo.org/record/34448.

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
