# Peer review of "Production of fatty acids in Ralstonia eutropha H16 by engineering β-oxidation and carbon storage"

_PeerJ, doi:10.7717/peerj.1468_

## Round 0.1 · original submission · Minor Revisions

The reviewers have some concerns, please address them.

·

Basic reporting

This manuscript described the engineering of R. eutropha for the production of fatty acid by blocking the beta-oxidation and carbon storage. The key of study lies on its identification q of three acyl-CoA ligase by RNA-se.

Experimental design

1. the authors engineered the fatty acids production in Ralstonia eutropha H16 by
blocking the β-oxidation and carbon storage. however, many studies have showed that overexpression and regulation of fatty acid de novo biosynthesis pathway have singnificant effecy on the production. Would author like to do that?
2. since engieering both beta-oxidation and carbon storge have effect on fatty acid production, why not combining the both oin one strain?
3. since different β-oxidation enzymes in R. eutropha may be induced by the presence of structurally distinct fatty acid substrates, the fatty acid uilization experiment of the different mutants should be done. this will also help to select the combiantion of different acyl-CoA ligase mutation.

Validity of the findings

no comments

Additional comments

no comments

Reviewer 2 ·

Basic reporting

The manuscript by Chen et al has engineered beta-oxidation pathway of Ralstonia eutropha to produce free fatty acids. The manuscrpt is written clearly and the results were fully discussed.

Experimental design

The experimental design was rational and well outlined.

Validity of the findings

The results of this study will be of interest to scientists working in the area of microbial production of fatty acids and fuels from bioresources.

Additional comments

This manuscript deals with engineered microorganisms for fatty acid production. It is a carefully planned and performed study. I recommend its publication in PeerJ. However, the format of temperature unit should be uniformed before publication.

Reviewer 3 ·

Basic reporting

No Comments

Experimental design

No Comments

Validity of the findings

No Comments

Additional comments

This manuscript described production of MCFA using engineered R. eutropha. Total fatty acid production was increased by knocking out PHB synthesis or knocking out the entry point for fatty acids into beta-oxidation. This study demonstrated that homologous beta-oxidation systems can be engineered to enhance fatty acid production. I recommend accepting with minor revision for quality improvement.

Fig 2A, it looks like the result of pJC025 is missing. I suggest labeling the data in the figure or mentioning in the figure legend. The fatty acid production increased as the TIR of RBS increased from 14,000 to 66,000, but a significant drop appeared when the TIR reached 84,000. The authors should discuss the reason.

Why the trend of fatty acid productivity of the strains harboring the 4 plasmids (pJC022 to 025) in Fig 3B was different as in Fig 2A, especially in WT and beta-oxidation mutant?

Knocking out either PHB synthesis or beta-oxidation pathway lead to enhancement of fatty acids production. Have the authors considered knocking out the two pathways simultaneously?

---

## Round 0.2 · Minor Revisions

A final minor revision is needed. The paper has reviewed some works on beta-oxidation engineering and carbon storages and so some of the important works on this area should be mentioned. For example, beta-oxidation pathway engineering has been reported as followings;

Chung A, et al. Microbial production of 3-hydroxydodecanoic acid by pha-operon and fadBA knockout mutant of Pseudomonas putida KT2442 harboring tesB gene. Appl Microbiol Biotechnol 83 (2009) 513 - 519

Liu Q, et al. Biosynthesis of Poly(3-hydroxydecanoate) and 3-Hydroxydodecanoate Dominating Polyhydroxyalkanoates by β-Oxidation Pathway Inhibited Pseudomonas putida. Metabolic Engineering 13 (2011) 11-17

Chung AL, et al. Biosynthesis and Characterization of Poly(3-hydroxydodecanoate) by ß-Oxidation Inhibited Mutant of Pseudomonas entomophila L48. Biomacromolecules 12 (2011) 3559–3566

---

## Round 0.3 · accepted · Accept

The revised paper is acceptable now.